# Differentially Expressed Genes, miRNAs and Network Models: A Strategy to Shed Light on Molecular Interactions Driving HNSCC Tumorigenesis

**DOI:** 10.3390/cancers15174420

**Published:** 2023-09-04

**Authors:** Saniya Arfin, Dhruv Kumar, Andrea Lomagno, Pietro Luigi Mauri, Dario Di Silvestre

**Affiliations:** 1School of Health Sciences and Technology, University of Petroleum and Energy Studies, Dehradun 248007, Uttrakhand, India; saniya.105605@stu.upes.ac.in (S.A.); dhruv.kumar@ddn.upes.ac.in (D.K.); 2Institute for Biomedical Technologies, National Research Council, F.lli Cervi 93, Segrate, 20054 Milan, Italy; andrea.lomagno@istitutotumori.mi.it (A.L.); pierluigi.mauri@itb.cnr.it (P.L.M.); 3IRCCS Foundation, Istituto Nazionale dei Tumori, Via Venezian, 1, 20133 Milan, Italy

**Keywords:** head and neck cancer, miRNAs, DEGs, PPI network, bipartite network, network topology, hubs

## Abstract

**Simple Summary:**

Head and neck squamous cell carcinoma (HNSCC) accounts for hundreds thousands deaths annually. We hereby propose a retrospective in silico study to shed light on gene–miRNA interactions and topological biomarkers driving the development of HNSCC. To achieve this, gene and miRNA profiles are holistically reevaluated using protein–protein interaction (PPI) and bipartite miRNA–target networks. The landscape of our findings depicts a concerted molecular action in activating genes promoting cell cycle and proliferation, and inactivating those suppressive. In this scenario, genes, including VEGFA, EMP1, PPL, KRAS, MET, TP53, MMPs and HOXs, and miRNAs, including mir-6728 and mir-99a, emerge as key players in the molecular interactions driving HNSCC tumorigenesis.

**Abstract:**

Head and neck squamous cell carcinoma (HNSCC) is among the most common cancer worldwide, accounting for hundreds thousands deaths annually. Unfortunately, most patients are diagnosed in an advanced stage and only a percentage respond favorably to therapies. To help fill this gap, we hereby propose a retrospective in silico study to shed light on gene–miRNA interactions driving the development of HNSCC. Moreover, to identify topological biomarkers as a source for designing new drugs. To achieve this, gene and miRNA profiles from patients and controls are holistically reevaluated using protein–protein interaction (PPI) and bipartite miRNA–target networks. Cytoskeletal remodeling, extracellular matrix (ECM), immune system, proteolysis, and energy metabolism have emerged as major functional modules involved in the pathogenesis of HNSCC. Of note, the landscape of our findings depicts a concerted molecular action in activating genes promoting cell cycle and proliferation, and inactivating those suppressive. In this scenario, genes, including VEGFA, EMP1, PPL, KRAS, MET, TP53, MMPs and HOXs, and miRNAs, including mir-6728 and mir-99a, emerge as key players in the molecular interactions driving HNSCC tumorigenesis. Despite the heterogeneity characterizing these HNSCC subtypes, and the limitations of a study pointing to relationships that could be context dependent, the overlap with previously published studies is encouraging. Hence, it supports further investigation for key molecules, both those already and not correlated to HNSCC.

## 1. Introduction

Head and neck squamous cell carcinoma (HNSCC) indicates a group of heterogeneous tumors that are derived from the squamous epithelium of the oropharynx, hypopharynx, oral cavity, and larynx. It is the 7th most common cancer worldwide, and it accounts for more than 325,000 deaths annually [1]. In approximately two thirds of the patients, it is diagnosed at the advanced stages, i.e., III/IV, and despite therapeutic advancement, the survival probability is considerably low [2].

The main treatment of HNSCC remains as surgery and radiotherapy with/without conventional therapy, including boron neutron capture therapy (BNCT), first approved in Japan in March 2020 [3], as well as sonodynamic therapy [4]. Platinum-based chemotherapy is the typical systemic treatment in recurrent metastases, but less than 20% of patients respond favorably [5]. This gap requires the discovery of new diagnostic and prognostic markers to give a boost for developing more specific and effective therapies. Recently, neoadjuvant immunotherapy has shown potential to improve clinical outcomes by increasing the antitumor immune response. In particular, EGFR targeting monoclonal antibody cetuximab was approved for the treatment of late-stage HNSCC, while the anti-programmed death-1 (PD-1) immune checkpoint inhibitors nivolumab and pembrolizumab were both approved for the treatment of patients with recurrent or metastatic HNSCC [6].

Epigenetic processes, including DNA methylation, histone modifications, and post-transcriptional gene downregulation following the action of non-coding RNAs, play a key role in the development and progression of HNSCC [7,8,9]. These changes regulate the accessibility of DNA to the cellular machinery responsible for transcription, replication, and repair, influencing various cellular processes. In this scenario, miRNAs have been demonstrated to have an impact on HNSCC, affecting its initiation, progression, metastasis, angiogenesis and resistance to therapeutic interventions [10]. Their marked variations in malignant tissues suggest the potential utility of miRNAs as standalone prognostic indicators [11]. Indeed, miRNA expression is specific to various tumor sites, and its analysis can help in determining clinical and pathological features, such as the degree of differentiation, or a predilection to metastasize [12].

Many miRNAs have been found to be differentially expressed in human cancers [13]. The impact of their dysregulation may be both tumor promoting and tumor suppressive [14]. In the context of oral cancer, epigenetic mechanisms, such as DNA hypermethylation, have been shown to disrupt the normal expression patterns of miRNAs [15]. Furthermore, an array of investigations have showcased a robust correlation between altered miRNA expression patterns and the principal risk factors implicated in the development of oral cancer, such as tobacco, alcohol, and viral infections [16,17]. For instance, the epigenetic silencing of miR-329 and miR-410 may be induced by arecoline, a carcinogenic ingredient of betel quid. It was associated with the upregulation of the Wnt-β-catenin pathway and the induction of expression of CCND1 and c-MYC, which plays a significant role in oral carcinogenesis [18]. Also, there is evidence that miRNAs themselves can act on epigenetic regulators, such as DNA methyltransferase, creating a complex feedback loop affecting the overall epigenetic landscape [17].

In addition to tobacco and alcohol, currently, the human papilloma virus (HPV) is considered another independent risk factor for HNSCC [19], mainly in oropharyngeal squamous head and neck cancer, where the prevalence of HPV ranges from 50 to 90% [20]. This infection is recognized to play a role in HNSCC pathogenesis, and HPV+ and HPV− patients show both clinically and biologically distinct features with reported genome-wide hypomethylation and promoter hypermethylation in HPV+ [21].

Since miRNAs play a significant role in clinical outcome, their presence in the blood of cancer patients may be used for monitoring the disease states, holding the potential to facilitate timely diagnosis and the design of targeted therapies [22]. The main upregulated miRNAs associated with HNSCC include miR-21, miR-455-5p, miR-155-5p, miR-372, miR-373, miR-29b, miR-1246, miR-196a, and miR-181, while those that are downregulated comprise miR-204, miR-101, miR-32, miR-20a, miR-16, miR-17 and miR-125b [23]. A study on nasopharyngeal carcinoma demonstrated that miR-9, miR-124, miR-892b, and miR-3676 were upregulated in plasma after drug treatment and downregulated at recurrence or metastasis, nominating them as potential markers for disease progression [24]. miR-31 has been suggested as a candidate for the early diagnosis of HNSCC since its low expression is correlated with tumor and lymph node metastasis [25]. Again, a study showed that the inhibition of miR-124 and miR-766 enhances the sensitivity of HNSCC cell lines to 5-fluorouracil and cisplatin (FP) chemotherapy and radiotherapy. In contrast, their overexpression confers resistance and increased cell invasion and migration [26].

In order to compose a molecular puzzle of HNSCC to be as complete as possible, several authors dedicated their efforts to characterizing differentially expressed genes (DEGs). A recent retrospective study based on bioinformatics analysis showed that the identified DEGs were mainly involved in drug metabolism cytochrome P450 and serotonergic synapses. Meanwhile, three key genes—CEACAM5, CEACAM6 and CLCA4—were significant for survival analysis [27]. In this landscape, the epidermal growth factor receptor (EGFR) emerged from several studies. Its overexpression was found in about 90% of HNSCC cases, and it has been correlated to poor prognosis along with radiation therapy resistance [28,29,30]. Several other genes, including PIK3CA, CDKN2A, NOTCH1, MET, CCND1, PIK3CA and TP53, have been strongly correlated with HNSCC. The latter, TP53, is a tumor suppressor gene, whose alterations have been observed in about 70–80% HNSCC patients [31]. They represent potential key targets for the design of therapies based on the miRNAs that regulate their expression [32], a strategy recently followed by some researchers for inhibiting EGFR by artificial miRNAs [33].

Based on these premises, we here propose a retrospective study based on RNAseq analysis of genes and miRNAs from HNSCC and control samples. The characterized profiles were re-evaluated by systems biology approaches based on graph theory [34]. These strategies were adopted in a few studies dedicated to HNSCC. Some authors relied on them to identify network signature useful for monitoring disease progression [35,36,37], while a few others aimed to identify potential therapeutic targets [38,39]. In our work, we investigated a cohort of 523 tissues from subjects affected by HNSCC at different stages. Differentially expressed genes (DEGs) and miRNAs were transformed in protein–protein interaction (PPI) and miRNA-target network models, which were subsequently evaluated at the functional and topological levels [40,41]. Our goal is to shed light on the molecular variations underpinning the different phenotypes and disease stages, as well as the identification of gene and miRNA hubs. These molecules could play the role of key molecules regulating processes, pathways and functions involved in tumor evolution mechanisms, thus potentially impacting patient survival.

## 2. Materials and Methods

### 2.1. RNAseq Data, from miRNAs to Differentially Expressed Genes (DEGs)

We accessed the TCGA Data Portal, where we retrieved the TCGA-HNSC project (ID phs000178, https://portal.gdc.cancer.gov/projects/TCGA-HNSC, accessed on 1 December 2022) data. The project reports bulk RNA sequencing for both gene and miRNA expression of 523 HNSCC tissue samples at different stages, and 44 paired control samples. In total, 21 patients were at stage I of tumor development, 97 at stage II, 106 at stage III and 278 at stage IV. Stage IV was split into stage IVa with 268 patients, and IVb counting 10 patients. Due to the low number of patients, stage IVb was not considered in our study.

RNAseq data used for this study include 561 RNAseq and 568 miRNASeq data files. To ensure the reliability of downstream analyses, the data were processed for quality control by filtering out low-quality or misaligned reads, using standardized data processing pipelines. The RNAseq sequencing reads were aligned to a reference genome (GRCh38) using the STAR (Spliced Transcripts Alignment to a Reference) alignment workflow, while miRNA sequencing was generated through the British Columbia Genome Sciences Center (BCGSC) workflow. Specifically, sequence read quality scores, adapter trimming, removal of low-quality reads and filtering out of any technical artifacts were performed. The aligned reads were quantified to determine the expression level of each mRNA and miRNA as normalized reads per million (RPM). Owing to a large sample size, we used the DESeq2 Bioconductor R package to analyze the RNAseq results and extract both differentially expressed mRNAs and miRNAs. DESeq2 accounts for compositional effects and library size differences employing the median-of-ratios normalization method. Further, it fits a generalized linear model (GLM) to count data and compare groups, and it employs a shrinkage estimation of dispersion [42].

The statistical significance of the differential gene expression between tumors and controls was assessed by the Wald test (adjP ≤ 0.01) [43]. To account for multiple hypothesis testing, DESeq2 applies a multiple testing correction procedure, the Benjamini–Hochberg one, which controls the false discovery rate (FDR). Statistical thresholds, such as adjusted *p*-value cutoff and fold change, were applied to determine significantly differentially expressed genes, which were visualized using volcano plots. Finally, a further list of differentially expressed genes and miRNAs was extracted by comparing patients stratified per clinical stage; in this case, the ANOVA test was applied (p≤0.01).

### 2.2. Extraction of miRNA Targets and Bipartite Network Reconstruction

Starting from differentially expressed miRNAs, we extracted the corresponding experimentally validated targets from miRecords [44], miRTarBase [45] and TarBase [46] database using an in houseR script. By using Cytoscape [47], they were used to build two bipartite miRNA target networks (for up- and downregulated miRNAs). These networks were processed by CentiScape Cytoscape’s App [48] to calculate the node degree centrality, rank miRNAs by the number of genes they target, and rank genes by the number of miRNAs targeting them. To select the miRNAs with the most target genes, as well as the gene target of multiple miRNAs, we used the average values calculated on the whole network as a threshold as previously reported [40].

### 2.3. Functional and Topological Analysis of PPI Networks Reconstructed from miRNA Targets and DEGs

Experimentally validated miRNA targets were used to reconstruct two different protein–protein interaction (PPI) networks using STRING Cytoscape’App [49]; only PPIs “experimental” or “database” annotated with a score higher than 0.1 and 0.3, respectively, were considered. To select the best candidate targets, we retained those with Degree ≥ 2∗AverageDegree (AverageDegree = 4 for up-regulated miRNA targets; AverageDegree = 5 for the downregulated miRNA targets). Genes targeted by both down- and upregulated miRNAs were filtered for significant difference in degree (upregulated miRNA target degree ≥ 8 AND downregulated miRNA target degree < 5, or upregulated miRNA target degree < 4 AND downregulated miRNA target degree ≥ 10). As a result, two networks of 415 (5856 edges) and 422 nodes (9423 edges) were reconstructed for up- and downregulated miRNAs, respectively.

A further PPI network model was reconstructed starting by DEGs found by comparing tumor and control samples (adjP ≤ 0.01; FC ≥ |2|). As reported above, the network was reconstructed by the STRING Cytoscape’App and only PPIs that are “experimental” or “database” annotated with a score higher than 0.1 and 0.3, respectively, were taken into consideration.

Using STRING Cytoscape’App, the reconstructed PPI networks were also functionally analyzed by retrieving the most enriched term from the GO Biological Processes, KEGG, Reactome and WikiPathways. The same networks were topologically analyzed through the centralities available in the CentiScape Cytoscape’s App [48]. The diameter, average distance, degree, betweenness, centroid, stress, eigenvector, bridging, eccentricity, closeness, radiality and edge centralities were calculated. Nodes with betweenness, centroid and bridging values above the average calculated from the whole reference network were considered hubs as previously reported [50]. The statistical significance of the topological results was tested by randomized network models [51]; n = 1000 random models per group were reconstructed and analyzed by in house R scripts based on VertexSort (to build random models), igraph (to compute centralities) and ggplot2 (to plot results) libraries.

### 2.4. Differentially Expressed miRNAs and Survival Analysis

The up- and downregulated miRNAs were tested for survival analysis. Special attention was paid to miRNAs targeting tumor suppressor genes (TP53, CDKN2A and PTPN4) and oncogenes (MET, VEGFA, KRAS, and MYCN), particularly if we found them relevant following RNAseq differential analysis and/or through the topological evaluation of the reconstructed network models. The relationship between miRNAs expression levels and patient survival outcomes was assessed using the log-rank test [52].

## 3. Results

### 3.1. Differentially Expressed Genes (DEGs) in Head and Neck Cancer

The comparison between HNSCC and control samples allowed the identification of 810 genes differentially expressed (DEGs) with adjP ≤ 0.01) and FC ≥ $\left|1.5\right|$ (Appendix A). A set of highly confident DEGs (FC ≥ $\left|2\right|$) was used for reconstructing a protein–protein interaction (PPI) network model and uncover the functional modules most affected by the disease state. DEGs were grouped in more than twenty functional modules, and the macro categories most represented were related to actin cytoskeleton, immune system, extracellular matrix (ECM), energy metabolism, proteolysis and cell cycle/proliferation (Figure 1).

The upregulation of specific gene families, such as metallopeptidases, collagens and homeobox proteins characterized the tumor samples. Actin cytoskeleton and ECM showed a clear opposite trend: ECM-related genes, including angiogenesis-related ones, were upregulated, while the expression of actin cytoskeleton-related genes, including cell adhesion and keratin ones, was lower than controls. As for cell cycle/proliferation, we noted the downregulation of genes involved in their inhibition, such as PSCA, NDRG2, SLURP1 and EMP1. Finally, at the immune system level, the upregulation of interferon signaling pathways emerged in HNSCC, while both glucose and lipid metabolism, with the exception of some apolipoproteins and fatty acid desaturases, were downregulated.

The topological analysis of the PPI network model (635 nodes and 4522 edges) reconstructed from all DEGs allowed the identification of 77 hubs (Table 1). In total, 54 out of 77 hubs were upregulated in the HNSCC group, while 24 were downregulated (Table S2). Among the top 20 ranked genes based on the betweenness centrality, we found AGRN, GNA12, FN1, PPL, TUBB3, CDKN2A, PDIA5, RECQL4, EPHB2, KIF4A, ALDH1A1, BGN, STAT1, CAV1, CLIC4, GSTA1, VCAN, CTSL, TJP3 and CRYAB (Figure 2a,b). Their trend of expression showed most of them to be upregulated in the tumor samples, regardless of the disease stage (Figure 2c), while only PPL, ALDH1A1, GSTA1, TJP3 and CRYAB were downregulated in HNSCC.

### 3.2. Differentially Expressed miRNAs in Head and Neck Cancer

Starting from the same cohort of profiles retrieved by GDC portal, we selected a set of differentially expressed miRNAs between HNSCC and control groups (Figure 3a). In detail, 38 and 34 miRNAs were up- and downregulated in HNSCC, respectively (Appendix A). Of note, upregulated miRNAs showed a differential expression and statistical relevance, increasing as the phase progresses. Meanwhile, it was less evident for downregulated ones (Figure 3b,c). As an example, miRNA-105-1, miRNA105-2 and miRNA-767 showed the higher fold change, and they were mainly upregulated at stage 3 and 4 (Figure 3b). Others, including miRNA-615 as best, were upregulated in all stages. On the other hand, miRNA-30a, miRNA-99a and miRNA-1258 were among those most confidently downregulated in all stages (Figure 3c).

Bipartite networks miRNA-target were reconstructed using up- and downregulated miRNAs as seed nodes. Their analysis had the purpose of identifying genes, pathways and biological processes potentially affected by the miRNA expression. Following node degree calculation, we ranked differentially expressed miRNAs based on the number of genes they target (Figure 4a). Of note, miR-301a, miR-301b and miR-615, upregulated in HNSCC, and miR-1, miR-101 and miR-29c, downregulated in HNSCC, were those with the highest number of known gene targets. In this scenario, the genes most targeted by downregulated miRNAs included VEGFA, G3BP2, PTBP1, IGF1R and DAZAP2. While, DEFB105B, PTPN4, ZNF711 and LDLR were those most targeted by upregulated ones (Figure 4b).

A further set of target genes were also differentially expressed. MET, FSCN1, COL1A1 and SLC16A1 were both upregulated in HNSCC, and targets of miRNAs were downregulated in HNSCC. On the contrary, RRAGD was both downregulated in HNSCC and the target of miRNAs were upregulated in HNSCC. In both cases, there is a correlation that suggests a direct relationship between the DEGs and miRNAs that target them. Furthermore, miRNAs downregulated in HNSCC had the greatest number of targets among genes upregulated in HNSCC (Kolmogorov–Smirnov test *p* ≤ 0.003), an observation that suggests the stronger regulation of this class of molecules in tumor development, than up-regulated miRNAs (Figure 4c).

### 3.3. PPI Network Models of Genes Most Targeted by Up- and Downregulated miRNAs: From Modulated Pathways to Hubs

Genes specifically targeted by up- and downregulated miRNAs were used to reconstruct two distinct PPI network models analyzed at functional and topological levels. Regarding the target genes of miRNAs downregulated in HNSCC, nine pathways were enriched. They include the VEGFA-VEGFR2 signaling pathway, MicroRNAs in cancer, and the hemostasis and relaxin signaling pathway. In contrast, genes targeted by miRNAs upregulated in HNSCC led to the enrichment of three pathways, including the transcriptional regulation by TP53, antigen processing: ubiquitination and proteasome degradation and iron uptake and transport (Figure 5a, Appendix A).

The topological analysis of the PPI network models reconstructed from targets of up- and downregulated miRNAs revealed relevant differences in terms of the centrality average values (Table 2). In particular, the target genes of downregulated miRNAs provided a PPI model with higher values of degree. In combination with lower values for diameter and average distance, these results describe a more compact and connected network. This is an observation that could fit the complex network of molecular interactions underpinning complex processes, like the cell cycle and its dysregulation. Indeed, as reported in (Figure 4c), downregulated miRNAs in tumors correlated with upregulated genes, suggesting a relationship which leads to their activation.

Following node centralities evaluation, we identified 36 and 38 hub genes in PPI network models from target genes of up- and downregulated miRNAs, respectively (Appendix A). Their selection was validated by the generation and processing of random networks, which showed a distribution of the average betweenness significantly different from the one obtained by processing the reference networks (Figure 5b,c). The best five ranked hubs targeted by up-regulated miRNAs included APP, ABL1, HSPA1B, LMBR1L and AAK1. On the other hand, KRAS, HNRNPH1, FLNA, XPO6 and TRA2B were the best five hubs linked to down-regulated miRNAs. Of note, FLNA and ENAH were both hubs and upregulated in HNSCC (Appendix A), increasing the likelihood that they could play a key role in tumor development.

### 3.4. Survival Analysis Using miRNAs Targeting Key Tumor Suppressors and Oncogenes

Starting from miRNAs differentially expressed by comparing tumor and control samples, we performed a survival analysis of HNSCC patients with the aim of identifying potentially exploitable candidates for drug development. Among all tested molecules, we selected a small set of up- (n = 3) and downregulated (n = 3) miRNAs.

As for miRNAs up-regulated, the best performance results were obtained for miR-301a, miR-3144 and miR-6728 (Figure 6a,b). Similarly, miR-29c, miR-378a and miR-99a were selected among downregulated miRNAs (Figure 6c,d). The best correlation between miRNAs expression and survival days was observed for miR-6728 (*p* = 0.012) and miR-99a (*p* = 0.0018). The other miRNAs showed probability values that are less meaningful. However, the expression of miRNAs upregulated (miR-301a, *p* = 0.053; miR-3144, *p* = 0.065) correlated with survival analysis better than downregulated ones (miR-29c, *p* = 0.114; miR-378a, *p* = 0.102). Noteworthy, miR-301a and miR-3144 include the tumor suppressor PTPN4 among their targets, while miR-29c and miR-378a include VEGFA. Furthermore, CDK6 and CCND2, important for cell cycle G1 phase progression and G1/S transition, are targets of miR-99a. On the contrary, TSC1, implicated as a tumor suppressor, is the target of miR-6728.

## 4. Discussion

The results shown in our study come from a comprehensive analysis of differentially expressed genes (DEGs) and miRNAs modulated in head and neck squamous cell carcinoma (HNSCC). The integration of these data with the protein–protein interaction (PPI) network models, as well as the reconstruction of bipartite miRNA-target networks, have proved useful in providing valuable insights into the molecular alterations associated with the disease state. Globally, the identification of more than eight hundred high-confidence differentially expressed genes emphasizes the substantial genomic modulation that occurs in HNSCC as highlighted in other studies [53,54]. Corresponding functional modules highlight the cytoskeletal regulation, immune system, extracellular matrix (ECM), energy metabolism and proteolysis as major players involved in the pathogenesis of HNSCC. The upregulation of ECM-related genes, including collagens and angiogenesis-related ones, suggest the tumor vascularization and aggressiveness [55]. Conversely, the downregulation of genes associated with the actin cytoskeleton and cell adhesion could be related to tumor cell motility and invasion, contributing to the metastatic potential of HNSCC [56], an hypothesis further strengthened by the specific upregulation of proteolytic proteins belonging to metalloproteases family [57,58].

Cell adhesion and motility significantly correlate with the decreased expression of EMP1 as previously reported [59]. Along with other cell-cycle inhibitors, such as PSCA, NDRG2 and SLURP1, they were downregulated in HNSCC, aligning with the hyper proliferative nature of cancer cells. The decreased expression of PSCA was reported by other authors [60], while more recently it was also described as a network hub [37]. Few investigations have put in evidence the role of NDGR2 in the context of HNSCC, but several studies proposed it as a tumor suppressor and metabolism-related gene in various cancers [61], and similarly, SLURP1, whose anti-proliferative activity is associated with nicotinic acetylcholine receptors, upon which it acts as an antagonist [62]. In contrast, in tumors emerged the upregulation of HOX genes encoding for proteins with a homeobox DNA-binding domain. Considered oncogenic biomarkers, this family was found to be overexpressed across a range of cancers, including HNSCC, where proliferation, migration and invasion are promoted [63,64].

The reduced expression of the tumor suppressor, combined with the upregulation of oncogenes, lets us imagine the concerted action of molecules, including miRNAs, in activating genes that promote the cell cycle and inactivating those that suppress it [65]. This panorama was complemented by the upregulation of cytokine and interferon signaling pathways. Although they could indicate an activated immune response, as a protective mechanism against tumor growth, recent studies reported that cancer-specific type-I interferon activation is associated with poor immunogenicity and worse clinical outcomes in HNSCC [66]. Moreover, cytokines induce alterations in the cellular and non-cellular tumor microenvironment, promoting ECM degradation, angiogenesis, immune evasion and metastasis [67].

In the perspective of gaining new insights concerning mechanisms regulating HNSCC hallmarks, the topological evaluation of network models contributed to selecting key molecules, such as gene and miRNA hubs. They are potential new markers in mediating communication and interactions within the HNSCC microenvironment. Among them, AGRN and PPL are hubs defined in other studies [68], remarking their relevance and, at the same time, the reliability of the approach here adopted. This was also confirmed by the topological relevance of genes that the literature has already associated with tumor development. As examples, we have FN1, whose over-expression correlates with the tumorigenesis, prognosis and radioresistance of HNSCC [69], EPHB2, that induces angiogenesis via the activation of ephrin reverse signaling [70], or CAV1, typically upregulated and related to the lipid metabolism [71]. In this context, the upregulation we observed for apolipoproteins (APOs) and fatty acid desaturase (FADS) gene families remarked the role of lipid profiles in HNSCC [72,73]. However, we did not appreciate this for the carbohydrate or glutamine metabolism, as reported in other studies [74].

Concerning new findings not yet clearly linked to HNSCC, some authors found that AGRN downregulation reduces cell proliferation, migration, invasion, and enhances apoptosis in colorectal cancer cells. Of note, they observed also that matrine, an alkaloid found in plants, has anti-tumor effects on colorectal cancer cells by inactivating the Wnt/β-catenin pathway via regulating AGRN expression [75]. Hence, AGRN could represent a potential target for therapies also in HNSCC. As for PPL, instead, we noticed that its hub role emerged in several studies [37,54,68]. Since it serves as a link between the cornified envelope, desmosomes and intermediate filaments, its downregulation could favor and enhance the invasive potential of HNSCC cells [76]. Similarly, the downregulation of ALDH1A1 and GSTA, protecting cells from reactive oxygen species, could describe a condition with reduced detoxification capacity and increased susceptibility to oxidative stress, which in turn promote tumor growth and survival [77,78].

The landscape painted through gene expression has improved with the exploration of molecules, such as miRNAs, involved in epigenetic regulation [7,8,9]. In addition to confirming miR-615 [79,80] miR-29c [80,81] and miR-101 [80] as signatures, the profiling of these elements contributes to dissecting the tangle of interactions underlying HNSCC tumorigenesis. In this view, bipartite networks evidence the upregulation of some genes, including MET [82], FSCN1 [83], COL1A1 [84], and SLC16A1 [85]. Being targets of downregulated miRNAs, these interactions gain a layer of relevance in promoting HNSCC pathogenesis.The same relevance was noticed for VEGFA and VEGFA-VEGFR2 signaling, i.e., the gene and pathway most targeted by miRNAs downregulated in HNSCC. These observations fit with a coordinated miRNAs action oriented to angiogenesis activation [86]. Thus, genes and miRNAs involved in these interactions represent an important target for anti-angiogenic therapies [87]. More generally, the observation that miRNAs downregulated in HNSCC have a higher number of targets among upregulated genes prospects a regulatory mechanism where these miRNAs might counteract the over-expression of oncogenic ones. Thus, they could represent a set of miRNA candidates to be investigated for therapeutic purposes.

In contrast to VEGFA-VEGFR2 signaling, the enrichment of transcriptional regulation by TP53, a tumor suppressor pathway, outlines the function of up-regulated miRNAs in shutting down processes controlling the cell cycle and proliferation [88]. This could fit also with the aim of inhibiting or escaping the immune system [89]; a plan supported by the high targeting of DEFB105B by upregulated miRNAs. In fact, it belongs to the defensin family that was for a long time considered to be merely antimicrobial peptides, but they can inflict DNA damage and the apoptosis of tumor cells, also attracting T cells, immature dendritic cells, monocytes and mast cells [90].

The epigenetic control of key genes involved in cell-cycle activation and inhibition was depicted also by the identification of miRNA-targeted hubs. ABL1, the best hubs targeted by upregulated miRNAs, is a protooncogene involved in many processes linked to cell growth and survival, including cytoskeleton remodeling, cell motility, DNA damage response and apoptosis. Of note, it has been also described to play a tumor-suppressor role in hematological malignancies [91]. In the same way, the KRAS gene stood out as the best hub targeted by down-regulated miRNAs. This relationship could lead to its upregulation in the tumor. As a matter of fact, KRAS is a well-known oncogene, whose mutations or overexpression are common in various cancers [92].

Although the epigenetic regulation of gene expression relies on different complex and interconnected mechanisms, the evidence of miRNAs targeting known tumor suppressors and oncogenes reveals potential interactions that may impact patient survival. Hence, they support the clinical relevance of miRNAs as potential therapeutic targets. Several authors addressed this topic by evaluating miRNAs in body fluids [93] and biopsies [94], or taking into account different molecules, like genes [95] and lncRNAs [96]. In our study, miR-99a provided the best result for survival analysis along with miR-6728. In addition to confirm the downregulation of miR-99a [97], our findings remarked its prognostic nature, already evidenced in other studies [98,99]. On the contrary, in the literature, there are not many reports that correlate the role of miR-6728 in the context of HNSCC [100], thus representing a potential novelty. Good results were obtained also for miR-301a [101], miR-3144, miR-29c [80,81], and miR-378a [102], targeting tumor suppressors or oncogenes emerged by our in silico analysis.

## 5. Conclusions

With this study, we wanted to contribute to a deeper understanding of HNSCC patho-biology, focusing on genes and miRNAs modulated in the disease stages. The integrated analysis through PPI network models has provided a comprehensive view of the molecular interactions correlated with HNSCC. In addition to presenting some potential novelties, our findings are in agreement with previous studies confirming the relevance of some mechanisms in HNSCC tumorigenesis and progression. In particular, the topological analysis highlights hub genes and miRNAs with a supposed key role in controlling HNSCC mechanisms. Thus, molecule candidates are potential drug targets. If, on one hand, this overlap is encouraging and supports further investigations, on the other hand, we cannot fail to take into account other molecules and mechanisms that would help complete the panorama of epigenetic regulation, including DNA and histone modifications, or lncRNAs, for which our knowledge is still in its infancy. In fact, the results presented support pathway regulation governed by a concerted action of multiple actors, both genes and miRNAs. And certainly the other epigenetic control mechanisms play a role that needs to be integrated in this framework. Despite some missing pieces, the characterization of specific miRNA–gene interactions, and their impact on patients survival, is another small brick supporting the development of prognostic and therapeutic strategies that ultimately aim to improve the clinical outcome of HNSCC treatments. Of course, we are well aware of the marked heterogeneity showed by the HNSCC subtypes, as well as of the numerous players and mechanisms affecting gene expression, limitations that are added to those of an in silico study focused on miRNA–target interactions, which in cancer could have context-dependent effects.

## Figures and Tables

**Figure 1 cancers-15-04420-f001:**
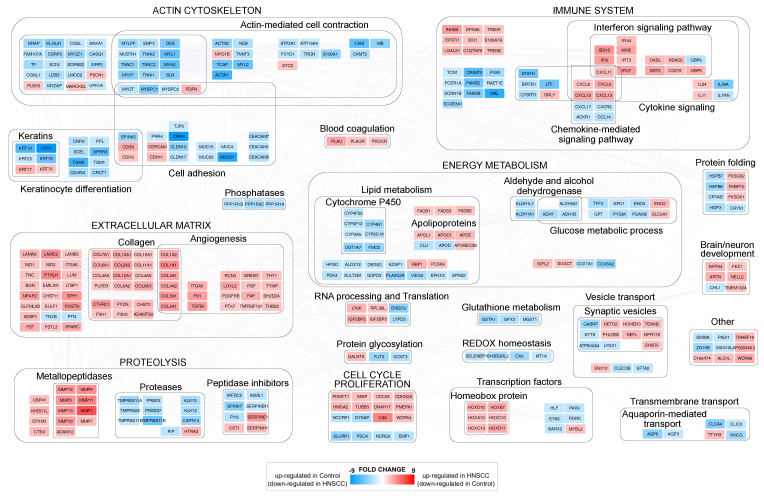
Protein–protein interaction functional modules differentially expressed by comparing HNSCC and control samples. In red (FC ≥ 2, adjP ≤ 0.01), genes upregulated in HNSCC (and downregulated in control), while in blue (FC ≤ −2, adjP ≤ 0.01), genes upregulated in control (and downregulated in HNSCC).

**Figure 2 cancers-15-04420-f002:**
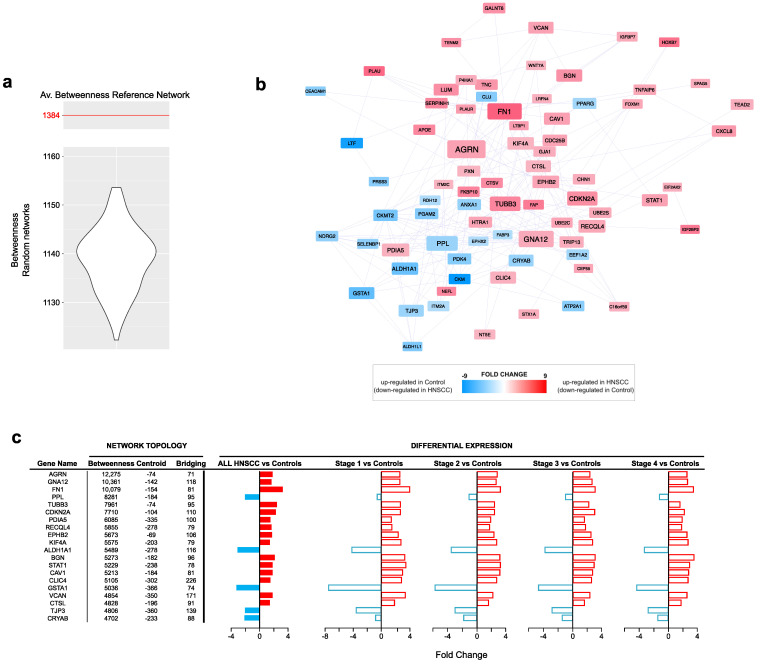
Protein–protein interaction network topology. (**a**) Violin plot of betweenness values from random networks (n = 1000) calculated from nodes (DEGs) in Reference Network. Compared to the reference network (Av. betweenness: 1384), random ones show significantly different average values and support the selection of hubs. (**b**) Hubs selected by betweenness, centroid and bridging centralities; all hubs had to show values higher than the average of the entire network. Red nodes indicate hubs upregulated in HNSCC (and downregulated in controls), while blue nodes indicate hubs upregulated in control (and downregulated in HNSCC). (**c**) Trend of expression, per disease stage, of the 20 best ranked hubs based on betweenness. Full red bars (FC ≥ 1.5, adjP ≤ 0.01) show the fold change of genes upregulated in HNSCC, while full blue bars (FC ≤ −1.5, adjP ≤ 0.01) show the fold change of genes upregulated in controls. For each gene, it is shown also the corresponding fold change at different stages (empty bars, adjP ≤ 0.01). Most hubs were upregulated in HNSCC and in all stages (red bars).

**Figure 3 cancers-15-04420-f003:**
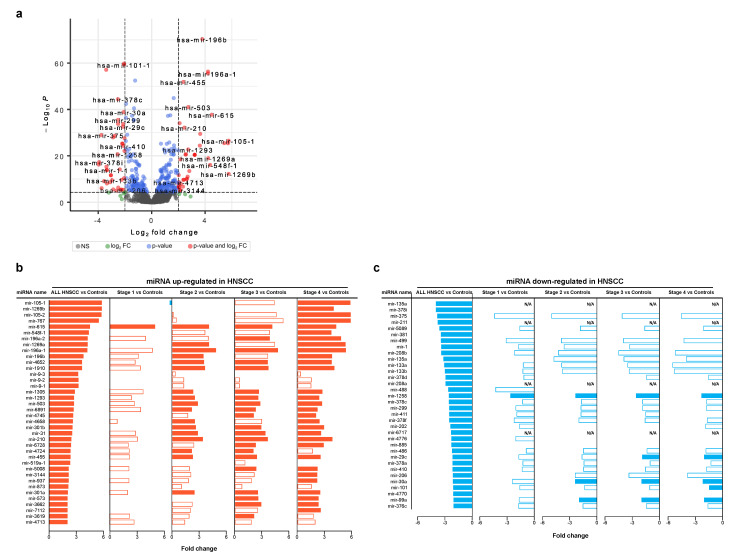
Differentially expressed miRNAs. (**a**) Volcano plot showing differentially expressed miRNAs by comparing HNSCC and control profiles (FC ≥ $\left|2\right|$ AND *p* values ≤ 0.00001). (**b**) Stage-level expression of miRNAs upregulated in HNSCC samples (*p* values ≤ 0.01). (**c**) Stage-level expression of miRNAs downregulated in HNSCC samples (*p* values ≤ 0.01). Full bars indicate the most significant differences (FC ≥ $\left|2\right|$ AND *p* values ≤ 0.00001).

**Figure 4 cancers-15-04420-f004:**
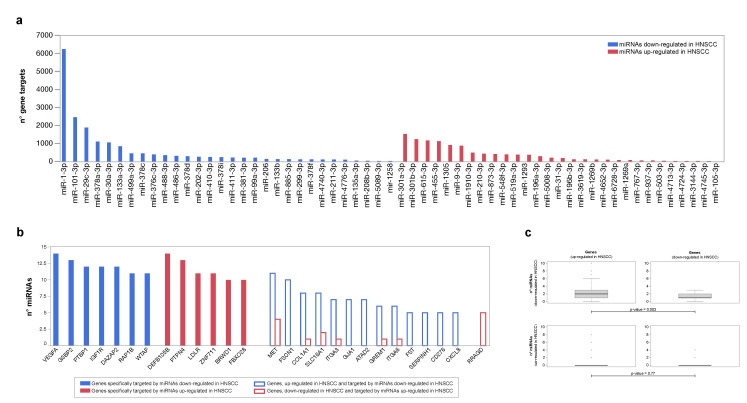
Differentially expressed miRNAs and corresponding target genes. (**a**) Up- and downregulated miRNAs ranked based on the number of genes they target. (**b**) Full bars indicate genes specifically targeted by up- and downregulated miRNAs and selected based on a Degree > 2∗Average Degree. Empty bars indicate differentially expressed genes (DEGs) most targeted by up- or downregulated miRNAs. (**c**) Differentially expressed genes (DEGs) in HNSCC, and number of miRNAs up- and down-regulated targeting them; Kolmogorov–Smirnov test (*p* ≤ 0.05).

**Figure 5 cancers-15-04420-f005:**
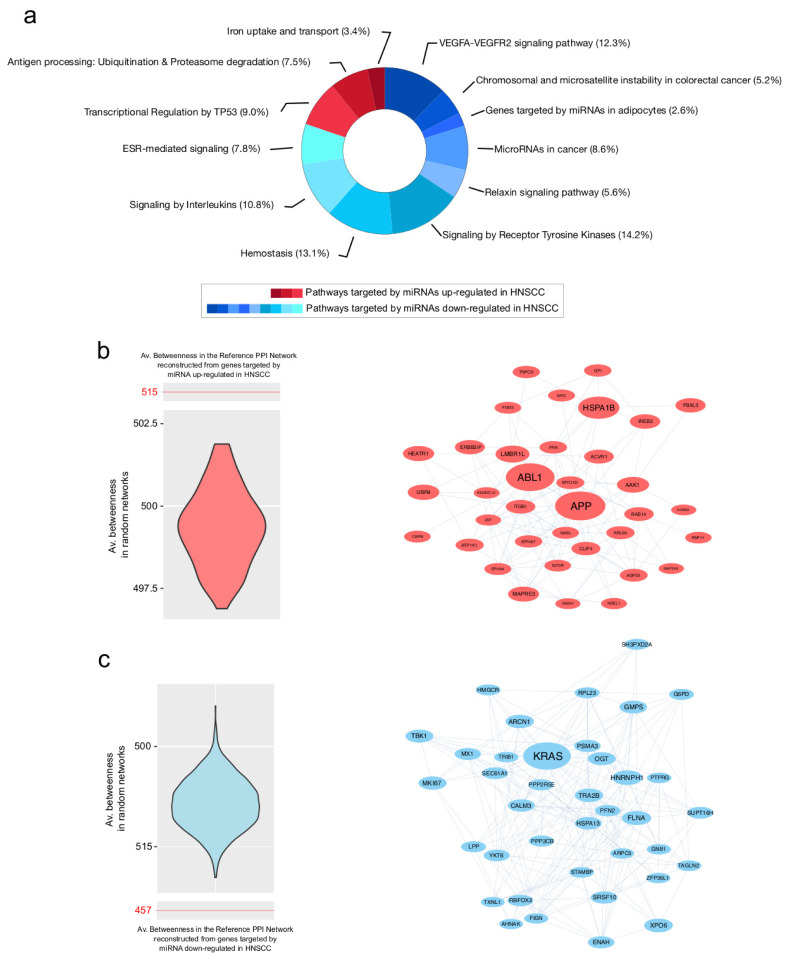
Pathways and PPI hubs from gene targets of up- and downregulated miRNAs. (**a**) Pathways enriched from gene targets of up- (red) and downregulated (blue) miRNAs in HNSCC. (**b**) On the left, violin plot of betweenness values from PPI random networks reconstructed from gene targets of miRNAs upregulated in HNSCC. Compared to the reference network (Av. betweenness: 515), random ones show significantly different average values and support the selection of hubs (on the right). (**c**) On the left, violin plot of average betweenness values from PPI random networks reconstructed from gene targets of miRNAs downregulated in HNSCC. Compared to the reference network (Av. betweenness: 457), random ones show significantly different average values and support the selection of hubs (on the right). All hubs were selected by betweenness, bridging and centroid centralities, which had to show values higher than the average of the entire network.

**Figure 6 cancers-15-04420-f006:**
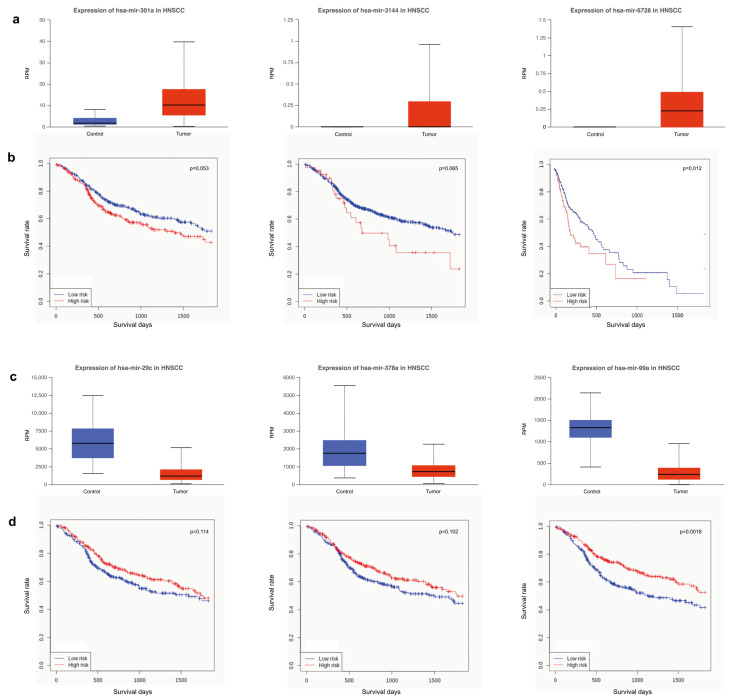
Differential miRNAs expression and survival analysis. (**a**) Expression level of miRNAs upregulated in tumor (miR-301a, miR-3144, miR-6728) and (**b**) survival analysis in HNSCC patients (Control, n = 44; Tumor, n = 482). (**c**) Expression level of miRNAs downregulated in tumor (miR-29c, miR-378a, miR-99a) and (**d**) survival analysis in HNSCC patients (Control, n = 44; Tumor, n = 482).

**Table 1 cancers-15-04420-t001:** Node and network centralities average values calculated by a PPI network models reconstructed starting from DEGs (FC ≥ $\left|1.5\right|$, adjP ≤ 0.01) by comparing HNSCC and control samples.

Centrality	Average Value
Betweenness	1384.6
Centroid	−437
Bridging	68.4
Degree	14.2
Radiality	4.8
Closeness	0.000506
Stress	18,601
Eccentricity	0.185
Eigenvector	0.0161
Diameter ^1^	7
Average Distance ^1^	3.18

^1^ Network centralities.

**Table 2 cancers-15-04420-t002:** Node and network centrality average values calculated by PPI network models reconstructed starting from genes targeted by up- or downregulated miRNAs.

Centrality	Average Value in PPI Network of Genes Targeted by Upregulated miRNAs	Average Value in PPI Network of Genes Targeted by Downregulated miRNAs
Betweenness	515.4	457.5
Centroid	−260.8	−234.5
Bridging	12.8	7.5
Degree	28.7	44.6
Radiality	3.75	2.91
Closeness	0.00109	0.00115
Stress	7883	9399
Eccentricity	0.274	0.309
Eigenvector	0.0353	0.0369
Diameter ^1^	5	4
Average Distance ^1^	2.24	2.08

^1^ Network centralities.

## Data Availability

Data for this study were retrieved accessing to the TCGA Data Portal and the TCGA-HNSC project, https://portal.gdc.cancer.gov/projects/TCGA-HNSC, ID phs000178 (accessed on 1 December 2022).

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
