# Peer review of "Differentially Expressed Genes, miRNAs and Network Models: A Strategy to Shed Light on Molecular Interactions Driving HNSCC Tumorigenesis"

_cancers, 2023, doi:10.3390/cancers15174420_

Round 1

Reviewer 1 Report

Dear,

This is an interesting article and authors investigated a retrospective in silico study to shed light on the molecular interactions driving the development of HNSCC and to identify novel topological biomarkers as a source for designing more specific and effective drugs. My decision is major revision.

-Abstract is ok, just summarize it a little more

-Introduction is poor and I suggest authors to explain more about epigenetic and oral cncer;

*Sonodynamic therapy and common head and neck cancers: In vitro and in vivo studies European Review for Medical and Pharmacological Sciences 2021 | Journal article DOI: 10.26355/eurrev_202108_26522 EID: 2-s2.0-85113912062 Part of ISSN: 22840729 11283602 CONTRIBUTORS: Hajmohammadi, E.; Molaei, T.; Mowlaei, S.H.; Alam, M.; Abbasi, K.; Khayatan, D.; Rahbar, M.;

*The Role of Epigenetic in Dental and Oral Regenerative Medicine by Different Types of Dental Stem Cells: A Comprehensive Overview Stem Cells International 2022-06-09 | Journal article DOI: 10.1155/2022/5304860 CONTRIBUTORS: Ahmed Hussain; Hamid Tebyaniyan; Danial Khayatan;

- Materials and Methods are well-explained

- Results are ok

-Discussion is poor and I suggest authors to use the above mentioned references.

Best,

Author Response

Q1. Abstract is ok, just summarize it a little more

A1. We thank the R1 for this positive comment. Following it, we tried to better summarize the Abstract.

Q2. Introduction is poor and I suggest authors to explain more about epigenetic and oral cncer;

*Sonodynamic therapy and common head and neck cancers: In vitro and in vivo studies European Review for Medical and Pharmacological Sciences 2021 | Journal article DOI: 10.26355/eurrev_202108_26522 EID: 2-s2.0-85113912062 Part of ISSN: 22840729 11283602 CONTRIBUTORS: Hajmohammadi, E.; Molaei, T.; Mowlaei, S.H.; Alam, M.; Abbasi, K.; Khayatan, D.; Rahbar, M.;

*The Role of Epigenetic in Dental and Oral Regenerative Medicine by Different Types of Dental Stem Cells: A Comprehensive Overview Stem Cells International 2022-06-09 | Journal article DOI: 10.1155/2022/5304860 CONTRIBUTORS: Ahmed Hussain; Hamid Tebyaniyan; Danial Khayatan.

A2. Following R1 comment we revised our Introduction citing also the manuscripts suggested.

Q3. Materials and Methods are well-explained

A3. We thank R1 for this positive evaluation.

Q4. Results are ok

A4. We thank R1 for this positive evaluation.

Q5. Discussion is poor and I suggest authors to use the above mentioned references.

A5. Following R1 comment we revised our Discussion.

Reviewer 2 Report

The title of manuscript is under question. English language has good quality. There are some explainations that are needed about the section "Results"

1. About figure 1

+ Please make this figure so clear that its texts can be easily read

2. About table 1

Why the average value of "Centroid" is -437?

3. About Figure 2

+ Please make this figure more obvious (specialy its texts)

4. Figure 3 have not enough quality to be in the manuscript because it contains some total vague information. Please make it bigger so that its text become readable

5. Please explain why the average values in PPI for "Centroid" are -260 and -234?

6. In page 11, line 266 the authors have mentioned that " our findings were in agreement with ... " but they have not mentioned some of these previous surveys. Please reform this part.

7. Line 268-270 needs proper reference

8. Please explain that what is the difference between your work and the manuscript below:

Molecular and genetic profile of head and neck squamous cell carcinoma

AI Stukan, VA Porhanov, VN Bodnya, O Yu Chuhraj, YM Makarova, IS Elizbaryan

Medical Herald of the South of Russia 9 (3), 50-57, 2018

9. Please check and adjust the "Reference list" based on the regulations of reference list of journal. (Titles, doi, the name of journal and ... )

Author Response

Q1. About figure 1, please make this figure so clear that its texts can be easily read

A1. We thank R2 for this suggestion. We are well aware that sometimes these maps are not easily readable. They show hundreds of linked genes and their expression to provide as systemic a view as possible. To improve its readability, we increased its size, we reduced the impact of the background interactions, and we increased the size of the tags showing the main enriched macro categories. We are confident that the figure in electronic format will be readable in every single gene because the figure provided is high resolution.

Q2. About table 1, why the average value of "Centroid" is -437?

A2. The centroid value suggests that a specific node has a central position within a graph region characterized by a high density of interacting nodes. Also here, "high" and "low" values are more meaningful when compared to the average centrality value of the graph G calculated by averaging the centrality values of all nodes in the graph. In biological terms, it can be interpreted as the "probability" of a protein to be functionally capable of organizing discrete protein clusters or modules. Thus, a protein with high centroid value, compared to the average centroid value of the network (as we reported in M&M), will be possibly involved in coordinating the activity of other highly connected proteins, altogether devoted to the regulation of a specific cell activity (Scardoni G, Tosadori G, Faizan M et al. Biological network analysis with CentiScaPe: centralities and experimental dataset integration. F1000Research 2015, 3:139(https://doi.org/10.12688/f1000research.4477.2))

The centroid value is computed by focusing the calculus on couples of nodes (v, w) and systematically counting the nodes that are closer (in term of shortest path) to v or to w. The calculus proceeds by comparing the node distance from other nodes with the distance of all other nodes from the others, such that a high centroid value indicates that a node v is much closer to other nodes. Thus, the centroid value provides a centrality index always weighted with the values of all other nodes in the graph. Indeed, the node with the highest centroid value (positive) is also the node with the highest number of neighbors. While all other nodes show negative values. For this reason, the average value of Centroid centrality is always negative.

Q3. About Figure 2, please make this figure more obvious (specialy its texts)

A3. Following R2 suggestion, we modified both style and text of the Figure 2 to improve its readability and understanding. In addition, we integrated the corresponding legend with major details.

Q4. Figure 3 have not enough quality to be in the manuscript because it contains some total vague information. Please make it bigger so that its text become readable

A4. Following R2 suggestion we modified our figure to improve its readability and understanding. Of note, as reported in our results, these graphs show how the differential expression of up-regulated miRNAs, as well as its statistical relevance, increases as the stage progresses, while it was less evident for down-regulated ones (L216-225).

Q5. Please explain why the average values in PPI for "Centroid" are -260 and -234?

A5. See point 2. In addition, the Centroid average value of a given network, like the average value of all Node centralities, has the function to be used as reference threshold to select nodes (in that network) with significant a Centroid (as we reported in M&M). On the contrary, it is not used to be compared with the Centroid average value of another network. While, it happens for Network centralies like Diameter or Average Distance.

Q6. In page 11, line 266 the authors have mentioned that " our findings were in agreement with ... " but they have not mentioned some of these previous surveys. Please reform this part.

A6. We thank R2 for this suggestion. We modified our Discussion properly.

Q7. Line 268-270 needs proper reference

A7. We thank R2 for this suggestion. We modified our Discussion properly.

Q8. Please explain that what is the difference between your work and the manuscript below: Molecular and genetic profile of head and neck squamous cell carcinoma AI Stukan, VA Porhanov, VN Bodnya, O Yu Chuhraj, YM Makarova, IS Elizbaryan. Medical Herald of the South of Russia 9 (3), 50-57, 2018

A8. The manuscript cited by R2 mainly focuses on the investigation of HNSCC in relation to the role of human papilloma virus (HPV) on cell cycle regulation, and specific molecules like Ñ€53 and Ñ€16. Our work, instead, is a retrospective study based on RNAseq profiles of genes and miRNAs from HNSCC patients and control samples. The novelty of our study concerns the re-evaluation of this dataset by systems biology approaches based on graph theory. So, our goal was to provide a systemic view to shed light on the molecular variations characterizing the different phenotypes and disease stages. We excreted gene and miRNA hubs as key molecules underlying processes, pathways and functions involved in tumor evolution mechanisms, thus potentially impacting on patient’s survival. As highlighted in our Introduction, these strategies have been adopted in few studies concerning HNSCC. In our work, we investigated a cohort of 523 tissue samples from subjects affected by HNSCC at different stages. Unfortunately, we did not find the manuscript cited by R2 in English language, and therefore it was not possible to make a more in-depth evaluation than that deduced from the abstract.

Q9. Please check and adjust the "Reference list" based on the regulations of reference list of journal. (Titles, doi, the name of journal and ... )

A9. We checked our Reference list. Only one paper, Okudela et al. 2013 International journal of clinical and experimental pathology 2013, 6, 1–12, is missing the DOI because not available. As for reference style we used the latex template provided by MDPI, thus, they are in agreement with MDPI guidelines.

Reviewer 3 Report

Differentially expressed genes, miRNAs and network models: a strategy to shed light on molecular interactions driving HNSCC tumorogenesis.

Saniya Arfin, Dhruv Kumar, Andrea Lomagno, Pierluigi Mauri and Dario Di Silvestre

See comments for authors:

In this study, authors have explored gene expression differences in Head and Neck Squamous Cell Carcinoma (HNSCC) versus controls, identifying 810 differentially expressed genes. They constructed a protein-protein interaction network to unveil functional modules affected by the disease. Notable categories included actin cytoskeleton, immune response, extracellular matrix, and metabolism. HNSCC showed up-regulation of angiogenesis-related ECM genes and down-regulation of actin cytoskeleton genes. They highlighted down-regulated cell cycle inhibitors (PSCA, NDRG2) and up-regulated gene families (metallopeptidases, collagens). Up-regulated interferon signaling and suppressed glucose/lipid metabolism were observed. miRNA expression revealed patterns linked to disease stage. Authors established miRNA-target networks, identifying key miRNAs and their target genes. Survival analysis indicated miR-6728-3p and miR-99a-3p as survival predictors, unveiling potential drug development candidates. Overall, the study emphasizes miRNA influence on HNSCC prognosis and offers therapeutic insights.

Minor comments:

There are grammatical mistakes that needs to be corrected. Many long sentences. Authors are advised to break long sentence to short and easily readable ones.

Suggestions for Rewriting:

·       Consider breaking down the section into smaller subsections to enhance readability and clarity.

·       Include brief statements highlighting the significance of the findings and their implications for cancer research and treatment.

·       Use bullet points or numbered lists for presenting specific findings, such as the number of DEGs, enriched pathways, and key miRNAs identified in survival analysis.

·       To avoid repetitiveness, use more varied sentence structures when summarizing similar findings, such as DEGs and miRNAs.

·       Consider using bold or italic formatting to emphasize key terms or findings within the comments.

Decision:

Accept after these comments/concerns have been addressed.

Quality of English is fine , however, some of the sentences can be broken down for better clarity.

Author Response

Q1. There are grammatical mistakes that needs to be corrected. Many long sentences. Authors are advised to break long sentence to short and easily readable ones. 

A1. We thank R3 for these comments. Following them we tried to improve our manuscript as suggested by R3.

Q2. Consider breaking down the section into smaller subsections to enhance readability and clarity.

A2. We thank R3 for this comment. Following it, we tried to globally improve our breaking down the section into smaller subsections.

Q3. Include brief statements highlighting the significance of the findings and their implications for cancer research and treatment.

A3. Following R3's suggestions, we have tried to improve our Discussion by better highlighting results and consequent implications. Furthermore, we have also created a Graphical Abstract to help readers understand the most relevant results.

Q4. Use bullet points or numbered lists for presenting specific findings, such as the number of DEGs, enriched pathways, and key miRNAs identified in survival analysis.

A4. As reported at point 3, we have now inserted a Graphical Abstract to help readers understand the most relevant results.

Q5. To avoid repetitiveness, use more varied sentence structures when summarizing similar findings, such as DEGs and miRNAs.

A5. We thank R3 for this suggestion. In the context of global revision of our manuscript we tried also to use more varied sentence structures

Q6. Consider using bold or italic formatting to emphasize key terms or findings within the comments.

A6. We don’t know if the use of bold or Italic fits we the journal style. However, as reported in previous points, we have now inserted a Graphical Abstract to help readers understand the most relevant results.

Round 2

Reviewer 1 Report

Dear,

There is no comment, my decision is acceptance. 

Reviewer 2 Report

No more comments.